# Overview of Biofluids and Flow Sensing Techniques Applied in Clinical Practice

**DOI:** 10.3390/s22186836

**Published:** 2022-09-09

**Authors:** Carlos Yáñez, Gerard DeMas-Giménez, Santiago Royo

**Affiliations:** Centre for Sensors, Instruments and Systems Development, Universitat Politècnica de Catalunya, 08222 Terrassa, Spain

**Keywords:** biosensors, body fluids, biofluids, clinical diagnosis, flowmetry

## Abstract

This review summarizes the current knowledge on biofluids and the main flow sensing techniques applied in healthcare today. Since the very beginning of the history of medicine, one of the most important assets for evaluating various human diseases has been the analysis of the conditions of the biofluids within the human body. Hence, extensive research on sensors intended to evaluate the flow of many of these fluids in different tissues and organs has been published and, indeed, continues to be published very frequently. The purpose of this review is to provide researchers interested in venturing into biofluid flow sensing with a concise description of the physiological characteristics of the most important body fluids that are likely to be altered by diverse medical conditions. Similarly, a reported compilation of well-established sensors and techniques currently applied in healthcare regarding flow sensing is aimed at serving as a starting point for understanding the theoretical principles involved in the existing methodologies, allowing researchers to determine the most suitable approach to adopt according to their own objectives in this broad field.

## 1. Introduction

Biosensors have been a prominent part of the successes that healthcare has reached since the beginning of what is sometimes referred to as “the modern age” to the present, mainly due to their valuable performance in measuring biological variables and promoting significant advances in clinical diagnostics, drug discovery, improvement in the quality of life of chronic patients, and health monitoring, among others [1,2,3,4,5,6,7,8,9,10,11,12,13]. Biosensors have been used in several medical devices, having a direct impact on both in vitro and in vivo applications [14]. With respect to the latter, their impact is manifold, as they excel at providing almost immediate and interactive information about the sample, allowing users to track, in real-time, the correct functioning of several vital variables in high-risk patients, thus facilitating 24 h monitoring [13,14,15,16,17,18,19].

In light of the above, the development of new biosensors to help clinicians measure and observe various aspects of the human body so that they can form a rapid and accurate diagnosis is a never-ending challenge. By its very nature, this is an endeavour that involves research from a wide variety of disciplines, from engineers and physicists to medical doctors and clinical physiologists, to name a few. These different profiles make it evident that a gap between technical and life sciences knowledge must be faced by every researcher (from an individual perspective) when the development of a state-of-the-art biosensor is intended.

In this literature review, we focus on well-established clinical methods for biofluid flow sensing, providing concise information on both technical and physiological subjects essential for fulfilling the previously mentioned gap, thus allowing researchers to understand the influence and characteristics of the main biofluids in the adequate functioning of human organs and systems, as well as the basic theoretical principles behind the functioning of several medical devices and techniques employed nowadays in healthcare for measuring the flow of such fluids.

We would like to emphasize that this paper is not intended to be a review of previous work, but rather a compendium of relevant topics for those interested in the development of flow sensors for biofluids within the healthcare field.

Since knowledge about the influence of biofluids on human health is of critical importance, reviews on the related topics have been previously published. Recently, [20] discussed the compartmentalization of total body water and described some methods by which the volume of the fluid compartments may be measured. On the other hand, [21] presented a literature review on the progress of clinical diagnosis and prognosis due to advances in vibrational spectroscopy techniques for biofluids; [22] published an evaluation of different body fluids and their advantages for the rapid detection of COVID-19, coupled with highly sensitive optical techniques for the detection of molecular biomarkers; other scientific literature regarding reviews focused on biofluids for biomarker research was also reported in [23,24,25,26]; an analysis of body fluids for forensic purposes was also presented in [27]; and, finally, overviews in different aspects of uncommon body fluids were issued in [28,29,30].

To the best of our knowledge, this is the first time that a concise but comprehensive review of the main body fluids influencing human health has been presented in the context of flowmetry endeavours, rounded off with a summarized description of the well-established techniques used nowadays for in vivo and in vitro flow sensing in healthcare settings.

Thus, the next section presents an overview of the physiological role of the principal body fluids in human health, with an emphasis on key aspects of the structure and functions of macrocirculation and microcirculation. We also describe the lymph and the lymphatic system in detail, as well as blood flow in the respiratory system, the digestive system, the brain, and the eye. The urinary system, the fluids within the gastrointestinal tract, cerebrospinal fluid, serous body fluids, synovial fluid, and other relevant human fluids are also concisely discussed. The third section of this review is devoted to explaining the principles behind some of the most important techniques for evaluating biofluids in healthcare. We observe the theoretical basis for angiography, lymphoscintigraphy, indocyanine green lymphography, plethysmography, photoplethysmography, particle imaging velocimetry, ultrasonic/acoustic flowmeters, laser Doppler flowmeters, the electromagnetic flowmeter, the nuclear magnetic resonance tissue blood flowmeter, positron emission tomography, lab-on-chip and organ-on-chip devices, and implantable sensors. A brief overview of some of the main future trends in the subject is also presented. In the fourth section, a discussion about possible concerns related to this review is developed and, finally, conclusions are drawn in the fifth section.

## 2. Overview of Human Biofluids

According to [31], the average water content of an adult man reported in the literature is from 58 to 80% of total body weight; moreover, the author himself found a percentage of 67.85. He also stated that the brain and heart are composed of ~73% water, the skin of ~65% water, the lungs of ~84% water, the kidneys of ~79% water, the liver of ~71% water, and the pancreas of ~73% of this vital liquid. All cells in nature have a certain amount of intracellular water and, in addition, they are immersed within an extracellular water compartment. It stands to reason, then, that most sensors designed for biological applications must consider, to some extent, fluid mechanic laws and the properties of fluids for an adequate design and function.

The intracellular compartment in the human body contains about 55% of the total water content and the extracellular compartment contains about 45% [32]. The extracellular space is divided into three additional compartments: interstitial space, intravascular space (or plasma), and transcellular space. The first accounts for nearly 45% of the extracellular water content, plasma for about 15%, and the transcellular space contains about 40%. The compartments of the human body are surrounded by a semipermeable membrane through which fluids pass from one space to another and which, at the same time, separates them [33]. Fluids normally move through the body due to various body conditions and physical forces.

Body fluids can be divided into categories such as saliva, cerebrospinal fluid, lymph, serous fluids, synovial fluid, bile, semen, vaginal secretions, human milk, amniotic fluid, urine, and blood, among others, depending on their variation in physical appearance, properties, cell types, and the number of cells [32].

There are several indicators of disease within alterations to the characteristics of body fluids, such as their colour, turbidity, number of cells per cubic millimetre, cell morphology, fluid volume, presence of abnormal cells/particles, and fluid velocity, to name a few. This section makes a brief and concise review of the anatomy and physiology of some of the main systems responsible for the distribution of fluids within the human body, paying special attention to the circulatory system and cutaneous blood flow due to their primary importance, as blood represents ~8% of the body mass of an average adult male [34].

### 2.1. Circulatory System

Simple unicellular organisms absorb the nutrients they need from their immediate environment, and the waste they produce is excreted into the same surroundings. In multicellular organisms, such as human beings, each cell acts as a unicellular entity, and its immediate environment is contained within the body. Organ cells exchange metabolic materials and waste via cell transport or diffusion, and the circulatory system is responsible for “circulating” these essential components throughout the body [35].

The circulatory system (also known as the cardiovascular system or the vascular system) circulates ~5 L of blood at a rate of ~5 L/m under normal resting conditions [36]. The circulatory system comprises the heart, which consists of two pulsatile pumps in series; the arteries, which carry blood and metabolic substrates from the heart; the capillaries, where the exchange of metabolic substrates and waste products with living tissues occurs; the veins, which carry deoxygenated blood; and the lymphatic vessels, which collect the extracellular fluid and return it to circulation.

The vascular network of the circulatory system is organized into complex structures in the form of hierarchal trees with varied branching configurations, designed to fulfil its equally complex tasks. Large arteries (>6 mm) are responsible for carrying oxygenated blood to smaller arteries (1–6 mm), which then source this supply to the arteriolar network (100–1000 μm) to finally reach the capillary beds (10–15 μm). Venules drain the deoxygenated blood from capillaries into larger veins, eventually reaching the heart [37]. Figure 1 illustrates the organization of a vascular tree.

The circulatory system is also essential in homeostatic tasks, such as the regulation of body temperature, the adjustment of the supply of oxygen and nutrients in different physiological situations, and humoral communication throughout the body [35].

There are three subsystems within the circulatory system [38]: (i) the systemic circulation, which receives oxygenated blood from the aorta and whose task is to divert it to the systemic capillaries. (ii) the pulmonary circulation, which is irrigated by the pulmonary artery and feeds the pulmonary capillaries; and (iii) the coronary circulation, which supplies the blood that the heart muscle (myocardium) needs to be able to supply blood to the rest of the body.

Although the coronary circulation is the smallest subsystem by blood volume, its proper function is obviously critical. According to [39], about one-third of the inhabitants of Western countries over the age of 35 years die from coronary artery disease; likewise, almost all elderly people experience some alteration of the coronary circulation.

The general path of the circulatory system begins in the left heart, which supplies oxygenated blood to the aorta at a relatively high pressure; the blood is then distributed to smaller arteries and then, finally, to the systemic capillaries, where oxygen is exchanged for waste carbon dioxide with the surrounding tissues. The deoxygenated blood containing waste products continues its path by flowing into the veins and, eventually, reaching the vena cava, which carries it to the right heart. From here, deoxygenated blood flows into the pulmonary artery, which progressively divides into smaller arteries, arterioles, and, eventually, into the pulmonary capillaries that surround the pulmonary alveoli, where the carbon dioxide carried in the haemoglobin of the red blood cells (RBCs) is exchanged for oxygen (which will also be carried in the haemoglobin) as part of the breathing process. Reoxygenated blood flows then from the lungs back to the left heart through the pulmonary veins, thus completing the cycle, as illustrated in Figure 2.

#### 2.1.1. Lymph and the Lymphatic System

The role of the lymphatic system in maintaining health is critical. In the heart, for example, cardiac oedema, inflammation, and fibrosis can be caused by a lymphatic blockage or dysfunction in the cardiac lymphatic vessels [40]. Therefore, in this subsection, we highlight this role by providing a detailed description of lymph and the lymphatic system.

Interstitial fluid (ISF) is produced by extracellular fluid generated in most tissues through a filtration process, which occurs when hydrostatic pressure in the capillary exceeds the opposite intravascular colloid osmotic pressure of plasma proteins [41]. ISF is a clear biofluid that contains solutes, proteins, lipids, waste substances from the metabolism of different tissues, and foreign materials from the interstitial spaces. As is illustrated in Figure 2, the lymphatic system is responsible for draining ISF from the interstitial spaces into the circulatory system. Once ISF enters a lymphatic capillary, it is called lymph [38].

According to [38], a volume of fluid equal to the total volume of plasma in a healthy human is filtered from the blood into the tissues each day; therefore, the lymphatic vessels collect, in normal conditions, ~3 L of ISF each day and return it to the venous system via lymph flow.

Like other extracellular fluids, lymph contains electrolytes and, in addition, a considerable amount of protein (40–60 g L^−1^), albumin being the main portion (23–34 g L^−1^). In peripheral lymph, the white blood cell count is ~5 × 10^8^ cells per litre, of which half are lymphocytes. On the other hand, lymph entering the thoracic duct contains ~1 × 10^10^ lymphocytes per litre, emphasizing the role of the lymphatic system in the immune response. Lymph drainage maintains the ISF protein level at ~15 g L^−1^, which is, normally, low enough to prevent oedema. ISF averages a colloid osmotic pressure of ~5 mmHg, ensuring a significant gradient between the interstitial tissues and capillary lumen, favouring tissue water reabsorption [42].

The lymphatic network is responsible for three main functions: (i) it maintains circulatory homeostasis by receiving ultrafiltrate from blood capillaries and an excess of protein from the fluid within the interstitial space, and then returns them to the venous system; (ii) lymphoid tissue in the intestinal tract absorbs digested fat, thus aiding in nutrition; (iii) the lymphatic system is a key player in defence, as bacteria, foreign antigens, and lymphocytes found in the interstitium drain with lymph to regional lymph nodes, activating the immune system [40].

Subcutaneous lymph-collecting vessels are commonly known as lymphatics and are found in almost all parts of the body that contain blood vessels. They begin in the superficial dermis but are not present in the epidermis [43], and they lie beneath the mucosa and serous lining of body cavities and major organs [40]. The junctions between the lymphatic cells are not tight junctions; overlapping cells form a freely permeable system that allows for the entry of ISF into the lumen of the lymphatic vessels. In the interstitial space, the lymph passes from the initial lymphatics to the lymphatic capillaries. Lymphatics are described as blind bulbs 5–50 mm wide [42], whose intimae are formed by elongated, single-layer endothelial cells supported by an elastic membrane. Their media are made up of transverse smooth muscle and fine elastic fibres, and the adventitia consists of connective tissue mixed with smooth muscle cells, fine blood vessels, and a fine nerve plexus [43].

Lymphatic capillaries are present in most body tissues and range from 20 to 70 μm in diameter. The palms of the hands, fingers, the soles of the feet, and toes are richly provided with such structures. These capillaries drain lymph to pre-collectors of up to 300 μm in diameter that contain valves and join together to form lymphatic vessels (see Figure 2), which are also valved [43]. The segments of vessels between the two valves are known as lymphangia. Smooth muscles in the walls of the lymphatic vessels cause the lymphangia to contract sequentially, allowing lymph to flow toward the thoracic region [42].

In the limbs, the superficial lymphatic vessels are located just below the skin, accompanying the superficial veins. Their perforators pass to the deep lymphatic vessels through the deep fascia. The deep lymphatic vessels are less numerous; however, they are larger than the superficial vessels and accompany the deep blood vessels [43].

Lymphatic vessels from the lower limbs enter the posterior abdominal wall after exiting the inguinal lymph nodes and, eventually, reach the cisterna chyli. From here, lymph enters the thoracic duct, which also receives lymph flow from the left side of the thorax, left arm, and the left side of the head and neck. The thoracic duct enters the venous system at the junction of the left subclavian vein and the left internal jugular vein. The right lymphatic duct drains lymph from the right side of the thorax, the right arm, and the right side of the head and neck [42].

Unlike the circulatory system, there are no active pumps to assist the lymphatic system; the movement of ISF from the surroundings of tissues through the lymphatic vessels is passive; a cycle of compression/relaxation causes lymphatic vessels to draw lymph. The pressure in the interstitial fluid around a capillary opens the overlapping cells of the lymphatic bulbs [38], and the endothelial array of the lymphatic capillary containing myoendothelial fibres contracts rhythmically, thus allowing the vessels to pump lymph toward larger lymphatic vessels, which, in turn, contain small bicuspid valves that ensure that the lymph is pumped in the right direction. Furthermore, the junctions between the endothelial cells act as a valve mechanism, which prevents lymph from returning to the interstitial space. After contraction, the connective tissue fibres that anchor the endothelial cells pull the lymphatic capillary back open, prepared to be refilled with more interstitial fluid [42].

Lymphatic pressures are only a few mmHg in the bulbs and smallest lymphatic vessels, but they can be as high as 25 mmHg in larger lymphatic vessels [42]. Lymphatic valves are responsible for this progression from low to high pressures [38].

At intervals along the lymphatic vessels, there exist lymph nodes. Lymph flows through the lymph nodes to reach the more proximal lymphatic vessels. Essentially, they consist of lymphoid follicles between 0.2 mm and 20 mm in length through which lymph is filtered [42]. There are about 500–600 lymph nodes clustered in the axilla, inguinal region, neck, pelvis, mediastinum, and para-aortic space [43].

A lymph node has a capsule leading to a trabecula, which divides the node into follicles containing sinusoids. Lymph nodes have a cortex into which the afferent lymphatics drain and a medulla containing lymphoid tissue that extends to the hilum, where the efferent lymphatics exit. Blood vessels enter and exit the hilum, but the arterioles, capillaries, and venules remain separate from the lymph that passes through the sinusoids within the lymph node [43].

As we mentioned before, lymph nodes facilitate the interaction between lymph and the immune system, being a key factor in the protection of the organism against infections and tumours [42].

#### 2.1.2. Macrocirculation and Microcirculation

Table 1 lists the average dimensions and approximate quantification of vessels in the human circulatory system. When blood flows through the aorta, which has an average diameter of 25 mm, the RBCs are small enough (~8 μm) for blood to be considered a homogeneous fluid. As the blood approaches the capillaries, which are similar in size to an RBC, a change between macrocirculation and microcirculation occurs. Capillary blood vessels are so narrow that their walls scrape the RBC membranes, so they flow in a single row. Although the frontier between these two subsystems is sometimes debated, the distinction between micro- and macrocirculation tends to be based on the Reynolds number and the Womersley number; if their value is much smaller than one, then the inertial force can be ignored, and the flow is considered microcirculation. Conversely, if both numbers are much greater than one, the fluid viscosity can be ignored and the flow is said to be macrocirculation. In the middle of this limit, the distinction becomes difficult to resolve and becomes inconsequential [44]. Under a different criterion, [38] microcirculation is defined as blood flow in vessels with a diameter of 100 μm or less. These microvessels include arterioles, metarterioles, capillaries, and venules. The authors of [38] also refer to the fact that almost all cells within the human body are close to a microvessel.

Microcirculation deals with ~10% of the circulating blood volume (in healthy conditions), and is the part of the circulatory system that is in direct contact with parenchymal cells [45]. Blood flow in microcirculation is usually referred to as perfusion, which is defined as the blood flow rate per unit of tissue volume. Its role in the exchange of water, hormones, nutrients, gases, and metabolic waste products between blood and cells is critical; in addition, microcirculation is responsible for regulating vascular resistance, the transport of heat, and the distribution of pharmaceuticals [38,46].

Capillaries are tubular structures with a diameter of 4–9 μm and one-cell-layer thick walls (~0.5 μm) made of highly permeable endothelial cells [38] (see Figure 1). As we mentioned before, filtration occurs when a fluid permeates the capillary walls and moves into the interstitial fluid of a tissue due to capillary hydrostatic pressure; the opposite process, that is, the movement of fluid from the interstitial fluid back into the capillaries, is called reabsorption, and it occurs due to the osmotic pressure (also referred to as oncotic pressure) caused by the difference in the solute-to-water concentrations in the blood and tissue fluid [46]. After leaving the capillaries, most blood returns to venules and later to the veins; however, ~10% of the fluid leaving the capillaries enters the lymphatic capillaries and returns to the blood through the lymphatic system [38], as is illustrated in Figure 2.

The study and knowledge of tissue hemodynamics are of great interest in many clinical areas that use it as a diagnostic tool for various pathologies related to the circulatory system. Perfusion is a key parameter in, for example, determining the viability of an organ or tissue before transplantation; the prognosis of patients suffering from ischemic heart disease; the determination of brain damage in stroke victims; and determining the condition of diseased organs such as kidneys, liver, pancreas, and lungs [47,48,49]. Perfusion is also considered in some drug studies, in cancer treatment, and in the laser photocoagulation of tumours. However, despite its mentioned clinical importance, there is no ideal way to measure perfusion to date [50,51].

#### 2.1.3. Histology of the Skin and Cutaneous Blood Flow

The easiest organ to assess through any device aimed to evaluate a biological flow profile is, obviously, the skin and its capillary network, since it is the outermost organ of the human body. In addition, the skin is the first barrier that a sensor must deal with before reaching any target of interest within the body. Since the angiogenesis process—the growth of new capillaries from pre-existing vessels—is a notable factor in cutaneous melanoma [52], one of the key uses of flow sensors is in the diagnosis and prognosis of skin cancer [53].

The integumentary system comprises the skin and its appendages (hair–hair follicles, nails, sudoriferous glands, and sebaceous glands), and is the external layer that covers the human body from head to toe, thus being the first protective barrier against harmful elements, such as fire, ultraviolet (UV) rays, dust, microbes, acids, etc. According to [54], for a 70 kg man, the extent of skin is ~1.7 m^2^ and its weight is ~3.86 kg, accounting for ~5.5% of the total body mass.

The skin is a stratified, heterogeneous, and anisotropic medium composed of two main layers joined together: the epidermis and dermis. Contrary to popular belief, the layer beneath the dermis, called the hypodermis or subcutaneous layer, is not considered part of the skin [55]. The thickness of each layer is different and, additionally, they are made of different solid materials saturated with fluids [56]. Their functions are different from each other, as each layer is seen as a system in itself. The main characteristics of these layers are described below.

##### Epidermis

The epidermis is the superficial layer of the skin, ranging in thickness from 0.07 to 0.12 mm along the surface of the body [57]. It is a keratinized, stratified squamous epithelium that is not innervated and is avascular; the nutrients that the epidermis requires arrive from the dermis by diffusion, and the removal of metabolic waste products also depends on the blood vessels of the dermis [55].

The epidermis is composed of up to five layers, which, viewed from the outside inwards, are: the stratum corneum, stratum lucidum (found only on palms and soles), stratum granulosum, stratum spinosum, and stratum basale. Collectively, the stratum spinosum and the stratum basale are usually referred to as stratum malpighi.

The epidermis contains four main types of cells: keratinocytes (90%); melanocytes (8%), which produce the pigment of the skin (melanin); Langerhans cells; and Merkel cells [55].

##### Dermis

The dermis is below the epidermis and above the hypodermis. Its thickness is between 1 and 4 mm and is composed of two layers: the papillary dermis and reticular dermis [57]. The main function of the dermis is to provide nutrients and physical support to the epidermis [55].

The papillary dermis is the upper layer, and it comprises about 10% of the entire dermal thickness. It contains thin collagen fibrils of 20–40 nm in diameter packed into thicker collagen fibres ranging from 0.3 to 3.0 mm in diameter [57]. In addition, the papillary dermis contains the nerves and capillaries that nurture the epidermis [55].

The reticular dermis lies below the papillary layer and is made up of strong connective tissue containing collagen and elastic fibres [55]; its collagen fibrils, of 60–100 nm in diameter, are composed, primarily, of type I collagen and are organized into fibres ranging from 10 to 40 mm in diameter [57].

In the average young adult, the collagen of the papillary dermis is similar to a randomly oriented fine fibre feltwork, whereas that of the reticular layer consists of large, wavy, and randomly oriented bundles of collagen that are loosely interwoven [57].

##### Dermal–Epidermal Junction

As explained in [58], the structural integrity of the skin depends on the dermal–epidermal junction, which is a complex network of extracellular matrix macromolecules that interconnect both layers.

The thickness of the dermal–epidermal junction is ~100 nm and is characterized by a wave-shaped pattern arising from the epidermal rete ridges to the papillary dermis; such elevations indent the epidermis, thus increasing the surface that is in contact with both layers, strengthening the dermal–epidermal connectivity and keeping, consequently, the dermis and epidermis layers firmly connected. Rete ridges surround the dermal papillae, which are small protruding extensions of the papillary dermis embedded within the epidermis. On a macroscopic level, fingerprints are a manifestation of this undulating pattern of ridges and furrows [58].

All the abovementioned layers of the skin are illustrated in Figure 3.

When the epidermis and dermis separate as a result of shear or friction forces, body fluids, such as lymph, serum, plasma, blood, or pus, can accumulate between the two layers, forming what is commonly known as a blister [55].

##### Vascularization of Human Skin

The skin receives vascular supply through two main networks of cutaneous arteries:(i)The superficial vascular plexus, which is a network of blood vessels located at the uppermost level of the dermis. From a histological point of view, this plexus marks the junction between the papillary and reticular dermis. The superficial plexus is composed of anastomosing small-calibre arterioles that branch off into capillaries, which extend into dermal papillae to supply the boundaries between the epidermis and dermis and envelop adnexal structures [60]. Each dermal papilla is provided with at least one capillary loop [58]. In Figure 4, an image of a pig skin sample—obtained through a 50× microscope objective and a lateral scan at different depths—shows an example of a capillary loop.

(ii)The deep vascular plexus, which is found at the joint between the dermis and hypodermis. It is a histological landmark that delimits the reticular part of the dermis from the hypodermis. This plexus is composed of medium-calibre vessels that emerge from larger vessels crossing the adipose septae of the hypodermis and is connected to the superficial vascular plexus by vertically oriented vessels [60]. Small tributaries sprout from this plexus to supply sweat glands, hair follicles, and other structures within the dermis [55].

Although the hypodermis is not considered part of the skin, the vascularization of this layer plays an important role in the blood supply of the skin because the subcutaneous arterial network has a larger extension and runs parallel to the cutaneous network, thus influencing the skin perfusion flow profile. Moreover, all the small arterial branches of the skin come from the hypodermis [61]. In Figure 5, the arterial network within the skin is illustrated.

#### 2.1.4. Blood Flow in the Respiratory System

The circulatory system and the respiratory system are strongly connected (see Figure 2), as the pulmonary circulation supplies blood to the alveoli for gas exchange. The bronchial system and the trachea, on the other hand, are fed by the bronchial circulation, which has high pressure and high oxygen content [64].

#### 2.1.5. Blood Flow in the Digestive System

The digestive tract provides the body with a constant supply of water, nutrients, and electrolytes. To do this, the gastrointestinal blood circulation distributes the absorbed substances through the gastrointestinal organs (e.g., oesophagus, diaphragm, stomach, spleen, liver, pancreas) and supplies essential oxygen for the absorptive and secretory functions of the gastrointestinal tissues, especially for those in the layer of the gut, such as the mucosa [64].

The blood vessels of the digestive system are known as the splanchnic circulation. After passing through the stomach, intestines, spleen, and pancreas, blood reaches the liver via the portal vein. Inside the liver, blood circulates through millions of sinusoidal capillaries, leaving the liver via the hepatic veins into the vena cava, which carries it back to the heart [64].

#### 2.1.6. Main Aspects of Blood Flow in the Brain and Eye

According to [64], the brain receives blood mainly from the left and right carotid arteries and, secondly, from the left and right vertebral arteries; the blood flow in the brain represents ~14–15% of total blood circulation. A branch of the carotid artery, the ophthalmic artery, enters the orbit of the eye through the optic canal and is divided into other branches that supply blood to the structures of the orbit, such as the lacrimal artery, which supplies the lacrimal gland, muscles, the anterior ciliary arteries, and the eyelid laterals; the long and short posterior ciliary arteries, which carry blood to the internal structures of the eyeball; the central retinal artery, which enters the optic nerve, dividing into many branches over the internal surface of the retina; the muscular arteries, which supply blood to the intrinsic muscles of the eyeball; the medial palpebral arteries, which feed the medial area of the lower and upper eyelids; the anterior ethmoidal artery, which supplies blood to the lateral wall and the nasal septum; the posterior ethmoidal artery, which delivers blood to the ethmoidal cells and nasal cavity; the dorsal nasal artery, which supplies blood to the upper surface of the nose; the supraorbital artery, which carries blood to the scalp and forehead; and the supratrochlear artery, which feeds the forehead.

The branches of the ophthalmic artery mentioned above, and the veins that drain the posterior part of the eyeball, contribute blood to the superior ophthalmic vein; on the other hand, the inferior ophthalmic vein receives tributaries from various muscles, as well as the eyeball posterior part [64].

### 2.2. The Urinary System

The circulatory system and blood flow are closely related to the urinary system; adequate blood volume and pressure are necessary for urine formation and, conversely, urine removes metabolic waste products, toxins, and drugs from the circulatory system [32].

The urinary system is made up of the kidneys, which are paired organs located in the lower back responsible for producing urine from the filtration of blood; the ureters, which transport the urine to the bladder; the bladder, where urine is stored; and the urethra, which connects the bladder with the external urethral orifice for urine excretion. For an average adult under normal conditions, daily urine production can range from 600 to 2000 mL/24 h. [32].

The important errands of the urinary system are the regulation of body fluids, maintaining the acid–base balance and the electrolyte balance, the maintenance of blood pressure and erythropoiesis and, of course, the excretion of waste products [32].

Urine contains several metabolites, such as glucose, cholesterol, urate, lactate, oxalate, and ascorbate, which can be analysed to diagnose various disease conditions, e.g., glucose found in urine is associated with diabetes [65,66], while the detection of lactate is used in the metabolic screening of inherited diseases [67,68,69].

### 2.3. Fluids within the Gastrointestinal Tract

There are several substances involved in digestion. Parietal cells secrete the glycoprotein gastric intrinsic factor, which is used in the absorption of vitamin B_12_. They also produce hydrochloric acid, which hydrolyses peptides and disaccharides and transforms the proenzyme pepsinogen into pepsin. In turn, pepsinogen is secreted by peptic cells and catalyses the degradation of proteins to proteases and peptones [32].

Other digestive enzymes secreted by the stomach are lipase, lactase, and peptidase. In addition, goblet cells and mucous glands secrete mucus that prevents the stomach walls from being damaged by acids and enzymatic activity [32].

### 2.4. Cerebrospinal Fluid

The capillary blood vessels within the ventricular choroid plexus produce ~70% of the cerebrospinal fluid by means of a combination of active secretion and plasma ultrafiltration. The remaining ~30% is formed by the cerebral/subarachnoid space and the ependymal lining cells of the ventricles. For a healthy adult, the volume of cerebrospinal fluid ranges from 90 to 150 m/L, with a production of 500 mL/24 h. The cerebrospinal fluid exits the ventricles through the foramina and circulates the hemispheres of the brain, descending over the spinal cord toward the nerve roots. The cerebrospinal fluid circulates slowly, allowing for prolonged contact with cells in the central nervous system. Later, in the dural sinuses, the cerebrospinal fluid is reabsorbed by the arachnoid villi [32].

The cerebrospinal fluid serves as a protective element, cushioning and lubricating the brain and vertebral column, avoiding injuries as a result of gravitational or inertial forces. It also exchanges nutrients and metabolic waste products with the brain and spinal cord [32].

### 2.5. Serous Body Fluids

This type of fluid, made up of ultrafiltrate plasma, is normally clear and slightly yellowish and is called serous because it resembles serum. The serous fluid is secreted by the serous membrane, which is a smooth tissue membrane lining the organs and walls of the different body cavities. The serous fluid fills the space between the visceral portion (the part that covers the organs) and the parietal portion (the part that lines the body walls) of the serous membrane, acting as a lubricant [32].

The serous cavities include the pericardium, which, in normal conditions, contains less than 50 mL of pericardial serous fluid surrounding the heart; the pleura, which normally contains less than 30 mL of pleural serous fluid around the lungs; and the peritoneum, which accumulates the peritoneal serous fluid (also known as ascites) that surrounds the organs contained within the abdominal cavity [32].

### 2.6. Synovial Fluid

Synovial fluid (also called synovia) is the viscous, mucinous substance that lubricates most joints; hence, its analysis is relevant for the diagnosis of the common diseases that affect human joints. Synovial fluid is produced in the synovium, a tissue that lines diarthrotic joints and allows bones to articulate freely [32].

Synovia is a dialysate of plasma that contains levels of glucose and uric acid equivalents to that of plasma and one-third of its protein levels. This dialysate is combined with a mucopolysaccharide synthesized by the synovium. Under normal conditions, synovia is a transparent and very viscous fluid, whose volume within the joints is small, e.g., the knee joint contains ~4 mL of synovia [32].

### 2.7. Other Relevant Human Fluids

This section concludes by briefly mentioning the main characteristics of semen, vaginal secretions, and amniotic fluid. Relevant information regarding less common biofluids can be consulted in [28,29,30].

Semen is made up of various fluids produced in different male organs. The seminal vesicles produce a slightly alkaline fluid that contains citric acid, fructose, potassium, and flavins. This fluid provides spermatozoa (which are formed in the sertoli cells of the seminiferous tubules of the testis) with these essential nutrients and comprises more than half the volume of semen. The prostate gland contributes 20% of the composition of semen, producing a fluid that contains citric acid, acid phosphatase, and proteolytic enzymes. The remaining portion of semen is produced in the urethral glands, bulbourethral glands, and the epididymis [32].

Vaginal secretions are produced in glands located on the cervix and their volume can vary throughout the menstrual cycle. Normally, vaginal secretions are clear mucus that can turn slightly white or pale yellow upon contact with air; abnormal changes in colour, amount, or consistency can be an indicator of different conditions or infections [32].

Amniotic fluid comes from the placenta and surrounds the developing foetus inside a membranous sac known as the amnion. In addition to being a protective element, it plays a key role in the exchange of water, nutrients, and biochemical products between the foetus and the maternal circulation. The composition of amniotic fluid is similar to that of the maternal plasma, with biochemical substances produced by the foetus and a low number of cells from the urinary tract, digestive tract, and skin of the newborn. When the foetal production of urine begins, urine also contributes to the amniotic fluid [32].

## 3. Current Techniques for Sensing Biofluids

As was mentioned before, this section presents a brief description of some of the well-established techniques applied in healthcare to assess the conditions of body fluids, which are useful in the diagnosis and prognosis of diverse diseases. Therefore, the information provided here allows us to delve further into the biomedical background necessary to understand the approaches that have already succeeded in the medical field, so as to identify either completely new approaches for gaps whose solutions are pending or opportunities for the improvement of existing technologies related to flow sensing.

Although it is impossible to list them all in a review, we selected, based on the specialized literature on the subject, the widest-used approaches.

### 3.1. Angiography

Angiography, also known as arteriography, is a well-established in vivo medical imaging technique used to visualize blood vessels. By means of a contrast dye injected into the vascular network through a catheter previously inserted into the femoral artery, an X-ray image makes it possible to visualize, in detail, the structures of blood vessels [36].

Digital subtraction angiography is a variant that allows the surrounding tissues and bones to be subtracted from the image, revealing only the vessels filled with the dye. Specific approaches are coronary angiography, which allows for the examination of the coronary arteries; cerebral angiography, which allows users to study arteries and veins within the brain; and peripheral angiography, which is useful in identifying stenosed vessels in limbs [36].

Due to the limitations inherent to the visual interpretation of simple 2D coronary X-ray angiograms, which make it difficult to assess coronary artery disease, several methods allowing for a more precise and objective 3D/3D + time diagnosis have been developed [70,71]. Recently, approaches such as magnetic resonance angiography and computed tomographic angiography have improved both spatial and temporal resolution such that high-quality cardiac imaging is now possible [71,72,73].

### 3.2. Lymphoscintigraphy

Lymphedema is a chronic and progressive disease that affects ~140–200 million people globally [74]. It is caused by the accumulation of lymph between cells due to the loss of drainage capacity in the lymphatic system, induced either by a blockage in the lymphatic tract or because the system itself is not growing [75].

Lymphoscintigraphy is a minimally invasive imaging technique that has largely replaced the more invasive and technically demanding contrast lymphangiography procedure in the evaluation of lymphatic function, as it is also associated with fewer complications and is easy to perform [76]. Because of this, it is recommended by the International Society of Lymphology [77] and is considered the benchmark for the diagnosis of lymphedema [78].

A gamma-emitting radioactive tracer, such as technetium 99m-labeled human serum albumin (^99^mTc-HSA) or dextran [40], is injected intradermally or subcutaneously in the lymphoedematous region. A gamma camera (also called a scintillation camera, an Anger camera, or, simply, a γ-camera) with a large field of view is then used to noninvasively image the transport of the radioactive tracer through the lymphatic system to determine if the lymphatic function is normal or pathological [76].

### 3.3. Indocyanine Green Lymphography

Indocyanine green (ICG) lymphography was first reported in 2007 for the evaluation of lymphedema and allows for a clearer visualization of superficial lymph flow than lymphoscintigraphy [78]. Because lymph flow can be visualized in real-time, ICG lymphography is a useful technique not only for lymphedema evaluation but also for studies prior to lymphatic surgeries.

ICG is a medical fluorescent green dye that is injected subcutaneously into the region of interest to allow for the identification of the superficial lymphatic tract beneath the skin [75], ICG lymphography is then performed using a near-infrared camera equipped with a 760 nm light-emitting diode and a filter-cutting light below 820 nm. The obtained fluorescent images allow for a real-time evaluation to a depth of up to 4 mm [79].

### 3.4. Plethysmography

Plethysmography is one of the most precise procedures for determining changes in blood volume in limbs, organs, and tissues. It includes the venous occlusion technique, which is a quantitative method for determining the tissue blood flow and has been the standard technique in healthcare for the measurement of tissue blood flow for decades [80], having outstanding performance when applied to limbs.

The venous occlusion method consists of two cuffs attached to the proximal and distal locations of a limb segment. At the distal cuff, a pressure greater than the maximum arterial pressure is applied to occlude the blood flow in all arteries and veins below the cuff. Next, a pressure slightly lower than the minimum arterial pressure is applied to the proximal cuff, resulting in an occlusion only in the veins at this level but not in the arterial blood flow. Finally, the volume change in the segment is measured, and the blood flow rate, *Q*, is determined by the rate of volume (*Vl*) increase [80]:(1)$$Q=\frac{dVl}{dt}.$$

In normal patients, the pressure applied to the proximal cuff is around 6.7 kPa (50 mm Hg), and the pressure applied to the distal cuff is around 20 kPa (150 mm Hg). In addition, the cuff pressure for venous occlusion should be applied for a very short time ([80] reports 0.1 s).

The principle of the venous occlusion technique applied to a segment of the lower leg is outlined in Figure 6. After the onset of venous occlusion, *Vl* increases due to arterial flow. When the occlusion is released, *Vl* returns to the pre-test condition. If the return time is slower than the normal value, there may be the possibility of a thrombus within the vessels in that segment of the limb [64].

Some variations in plethysmography are: displacement plethysmography (used in the diagnosis of lymphedema [81]), which uses either water- or air-filled containers to measure the change in the volume in the limb based on the variation in the level of the fluid (they are also known as fluid displacement plethysmographies and include mercury strain-gauge plethysmography and capacitance plethysmography); impedance plethysmography, where the change in the volume is measured by the change in electrical impedance; lung plethysmography, which measures respiration based on the volume change in the lung and includes body plethysmography, inductance plethysmography, and impedance pneumography [80]; and video plethysmography, where a video camera registers the colour variations on the surface of the skin on parts of the body such as face and hands, allowing for the extraction of the beat-to-beat pulsatile signal caused by the arterial pulsations in the blood flow [82].

### 3.5. Photoplethysmography

Photoplethysmography (also known as PPG), is a simple and inexpensive optical method used to monitor heart rate and pulse oximetry readings in healthcare. Recently, it has also been adapted into common-use wearables due to its straightforward implementation [83,84,85].

According to [85], PPG uses an infrared light source—usually an infrared light-emitting diode (IR-LED)—that emits toward tissue and a photodetector that measures changes in light intensity via reflection or transmission through the tissue. Such changes are proportional to the volumetric variations in blood circulation due, mainly, to the cardiac cycle, since the arteries contain more blood volume during systole than during diastole.

In reflectance mode, the photodetector detects the backscattered light reflected from structures within the illuminated area, such as tissue, bone, and blood vessels, as shown in Figure 7a. On the other hand, in the transmission configuration, a photodetector placed in front of the IR-LED detects the light transmitted through the medium, as illustrated in Figure 7b [85].

### 3.6. Particle Imaging Velocimetry

Particle imaging velocimetry (PIV) is an in vitro technique for tracking the movement of reflective particles through a flow channel. There are several PIV techniques; the most basic and popular one is simply called particle tracking. It consists of a camera (usually a high-speed camera) that records the displacement of particles immersed in a moving fluid as they pass through a very specific cross-section within a flow pipe to determine the characteristics of the flow by tracking individual particles at different points over time. Using computer algorithms, the velocity of the particles can be calculated from the change from a known position in the previous frames with respect to the new position in each subsequent frame, divided by the framerate of the camera. To facilitate tracking, reflective beads are commonly used and a high-intensity light source (e.g., a laser) illuminates the region of interest [36]. A schematic of a typical PIV configuration is shown in Figure 8.

It is also common to use more than one camera to collect multiple views of the same cross-section so that a complete 3D velocity flow profile can be obtained by cross-correlating the data (also called triangulation). Furthermore, in such configurations, the flow patterns can also be easily determined [36].

Some shortcomings in this method arise when the particles are too concentrated within the flow, so the reflected light can be outside the resolution limit of the camera, preventing the algorithm from discriminating between particles that are close to each other. In addition, if some particles do not follow the expected trajectory (e.g., particles that have erratic motion), the algorithm will not be able to correctly determine the velocity of such particles [36].

PIV is useful for testing new designs of cardiovascular devices (e.g., a stent, a mechanical heart valve, or a total artificial heart) to determine, whether and to what extent, they adversely affect the blood flow [36].

Other PIV techniques include laser speckle velocimetry, which uses the speckle pattern of particles suspended within the fluid that occurs when they are illuminated with a coherent source rather than the reflective pattern used in regular particle tracking. This variant is typically used to model the movement of large solids, as in simulations of venous thromboembolisms; holographic PIV, which is used to obtain 3D velocity profiles by collecting information about the locations of particles in a hologram, allowing users to reconstruct an image through a computational process; and a more recent PIV approach that uses fluorescent particles placed in a microflow stream and an epi-fluorescent microscope to investigate flow in artificial microcapillaries [36].

### 3.7. Ultrasonic/Acoustic Flowmeters

The ultrasonic/acoustic flowmeter allows physicians to measure flow instantly. By means of an ultrasonic probe, ultrasound waves (frequencies above 20 kHz [86]) are created and transmitted through living tissues, which allows them to analyse biocharacters such as blood flow or the blood profile [64].

A transducer converts electrical signals into acoustic waves; since transducers have a fixed diameter, they can produce diffraction patterns just as an aperture does in optics. A difference in diameter and shape thus provides near and far sensing capabilities, as illustrated in Figure 9 [64].

Theoretically, the best results occur within the near field or at initial distances, such that:(2)$$d_{nf}=\frac{D^{2}}{4\lambda }.$$
where *d_nt_* represents the near field distance, *D* is the diameter of the transducer, and *λ* is the wavelength.

The beam divergence angle for the far field, *θ*, can be found with:(3)$$\sin\theta =\frac{1.2\lambda }{D}.$$

Acoustic/ultrasonic flowmeters are widely applied in several industries; in medicine, commercial devices can measure blood flow in microvessels, and examples of such apparatuses can be found in [87]. This type of flowmeter can employ clamp-on and wetted transducers, single and multiple paths, paths on and off the diameter, contrapropagating transmission, active and passive principles, tag correlation, reflection (Doppler), liquid level sensing of open channel flow or flow in partially-full conduits, vortex shedding, and other interactions. Moreover, ultrasonic flowmeters can be used with gases, liquids, and multiphase mixtures [88]. In conventional blood flow ultrasound imaging, the frequencies used are typically in the range of 2 MHz to 15 MHz [86].

Examples of ultrasonic instrumentation for measuring blood flow are transit-time flowmeters and continuous-wave Doppler flowmeters.

#### 3.7.1. Transit-Time Flowmeters

According to [64], two different arrangements for transit-time flowmeters can be described; in the first approach, two transducers are placed diagonally to each other on both sides of a blood vessel or an artificial channel, as depicted in Figure 10a. In the second approach, two transducers are located on the same side of the channel, and a reflector is placed on the opposite side in the middle so that it reflects the waves between the transducers, as illustrated in Figure 10b.

The transit-time of the sonic wave travelling from one transducer to another is recorded, and the difference between the integrated upstream and downstream transit times makes it possible to measure the blood flow rate and the velocity of the stream inside the channel [64].

#### 3.7.2. Continuous-Wave Doppler Flowmeter

This kind of flowmeter emits an ultrasonic signal through the wall of the blood vessel or the artificial channel. The signal is then reflected by blood particles, such as RBCs, and reaches the receiver, as shown in Figure 11.

A difference between the sending and receiving frequency of the wave is registered due to the Doppler frequency shift caused by the moving RBCs, which allows us to calculate the average flow velocity as:(4)$$\frac{f_{D}}{f_{0}}=\frac{V}{c_{s}}.$$
where *f_D_* is the Doppler frequency, *f*_0_ is the source frequency, *V* is the target velocity, and *c_s_* stands for the speed of sound.

### 3.8. Laser Doppler Flowmeters

A laser Doppler flowmeter (LDF) allows for the continuous (or quasi-continuous) flow sensing of biofluids within a given tissue or inside artificial microchannels. It is a non-invasive method that uses the Doppler shift caused by moving particles (present naturally or artificially in a body fluid) to the photons of a laser beam, similar to the effect explained above for the continuous-wave Doppler flowmeter. This method has been adopted in dermatology and plastic and gastrointestinal surgery, mainly to assess the blood flow in tissues under clinical intervention. In addition, LDFs have been applied in spontaneous rhythmical variation studies to quantify fluctuations in skin blood flow [89] and coupled in a fundus camera to measure the relative blood velocity in the human optic nerve head [90,91], to name a few clinical applications.

Part of the coherent emission of the laser source, focused over the tissue under study, will reflect back from the surface owing to specular reflections. The remaining light penetrates the tissue and, because biological tissues are highly scattering and absorbing media, photons will undergo scattering and absorption events [80].

As tissues are composed of molecules containing discrete electrical charges, protons, and electrons, if they are irradiated by an electromagnetic wave, the electrical charges of the molecules will enter into an oscillatory motion state, radiating electromagnetic energy in all directions. This secondary reradiated light is known as scattered radiation [80].

Scattering occurs due to differences in the refractive index between the different elements that compose a given medium, and the direction of the scattered radiation largely depends on the size and shape of the scattering particles. For particles smaller than the wavelength of the beam that illuminates them, Rayleigh scattering occurs. Under this regime, the phase differences between the secondary electromagnetic waves leaving the irradiated particles are small, resulting, hence, in a homogeneous zone of light that surrounds each particle. On the contrary, when the dimensions of the particles are larger compared to the incident wavelength, the scattered radiation follows, more or less, the same direction as the incident beam. This is known as the Mie scattering regime [80].

Biological tissues are extremely heterogeneous media, having structures of different sizes and orientations; moreover, the scattered radiation from one particle will interact, logically, with other particles in the vicinity (the so-called multiple scattering effect); as a consequence, the resulting scattered field registered by a photodetector is very complex [92].

In the well-established configuration of an LDF, the light from the laser source is delivered to the tissue through a fibreoptic probe and is collected after being back-scattered, either by the same probe or by a secondary probe connected to a photodetector [86].

Considering the incident laser beam to be normal compared to the flow velocity, the Doppler frequency, *f_D_*, is provided by [86]:(5)$$f_{D}=\frac{VSin\theta }{\lambda }.$$

Here, *V* stands for the flow velocity, *λ* is the wavelength of the laser source, and *θ* is the angle between the incident and the scattered field.

### 3.9. Electromagnetic Flowmeter

The principle of operation behind the electromagnetic flowmeter (EFM) is the law of electromagnetic induction promulgated by Faraday, which states that a voltage is induced in a circuit when relative motion occurs between an electrical conductor and a magnetic field [93]. In the case of a fluid containing electrically charged particles flowing through a magnetic field, an electromagnetic force is produced. If one of these particles with a charge, *q*, moves with a velocity, *V*, within the magnetic field of the magnetic flux density, *B*, then a force, *F*, will apply to the particle in such a way that [80]:(6)F=q(V×B).

In the case of blood as an application example, since blood is an electrolytic solution that contains ions of positive and negative charges, when it flows through a magnetic field, the different charges move in opposite directions, thus producing an electric field, *E*, so that *F* is balanced with the electric force, *qE*:(7)qE+q(V×B)=0.

By placing two electrodes along this electric field, an electromagnetic field with a potential difference, *U* (in volts), can be registered as:(8)U=Y·E=−Y·(V×B).

Here, *Y* stands for a vector connecting the locations of the two electrodes placed perpendicular to the velocity vector with a distance, *d*, from each other, as shown in Figure 12. If *V* and *B* are also perpendicular, then:(9)U=VBd
where *d* is, logically, equal to the channel diameter.

The EFM principle is illustrated in Figure 12.

As stated in [80], the velocity at which blood flows inside a vessel is not uniform. Thus, in Equation (8), *V* represents the mean velocity as long as the velocity profile is axisymmetric with respect to the longitudinal axis of the channel. Then, the flow rate, *Q*, can be expressed as:(10)$$Q=\frac{\pi d^{2}V}{4}=\frac{\pi dU}{4B}.$$

The most common types of EFMs for blood flow sensing are sine wave EFM and square wave EFM; likewise, the use of these devices in clinical diagnosis is widely spread because they are one of the most accurate techniques for blood sensing, both non-invasively and invasively [94].

### 3.10. Nuclear Magnetic Resonance Tissue Flowmeter

Nuclear magnetic resonance tissue flowmeters are based on the principle of magnetic resonance imaging (MRI) [64]. When a nucleus that has a magnetic moment is placed inside a static magnetic field and is disturbed by an oscillating field of a specific frequency, it will respond by producing an electromagnetic signal with a characteristic frequency. The frequency at which this emission of magnetic energy occurs is known as resonant frequency, and it depends on the intensity of the static field and the type of nucleus [80].

The magnetic resonance of the ^1^H hydrogen nucleus, present in biofluids mainly in water molecules, makes it possible to measure the flow. Under a static field, *Z*, the resonant frequency, ν, is provided by [80]:(11)$$\nu =\frac{\gamma Z}{2\pi }.$$
where γ is the gyromagnetic ratio, which depends on the nucleus. For ^1^H, ν = 42.5 MHz when *Z* = 1 T [80].

Without the influence of the excitation field, the nuclei with a magnetic moment align along the direction of the static magnetic field, thus producing a magnetization, *M*, proportional to the intensity of the magnetic field, *Z*, such that:(12)$$M=\chi_{m}Z.$$

Here, $\chi_{m}$ is the magnetic susceptibility; e.g., in blood at 37 °C, the $\chi_{m}$ of ^1^H is 3.229 × 10^−9^ [80].

By applying an excitation field perpendicular to the direction of the static field at the appropriate resonance frequency, the precessional motion of the magnetic moment of the nuclei is induced, producing an oscillating magnetic field that can be recorded by different types of sensors, such as a pickup coil [80], as illustrated in Figure 13.

There are different MRI-based techniques for tissue flow measurement and perfusion imaging, such as the continuous wave excitation method, the time-of-flight method, and the phase contrast method.

In addition, magnetic resonance lymphangiography has recently emerged as an effective imaging modality for the lymphatic system [95].

### 3.11. Positron Emission Tomography

Positron emission tomography (PET) is a type of nuclear medicine procedure intended to obtain cross-sectional images of the human body in vivo. After a very low amount of a short-lived radioactive tracer (or a radiopharmaceutical) used as a labelling agent is injected intravenously, its concentration and decay within the tissue are measured by a PET scanner, as depicted in Figure 14.

As the radioisotope decays by positron emission (also called positive β-decay), it emits a positron, which has a charge opposite to that of an electron. After travelling a short distance, the positron meets an electron and they annihilate each other, producing two 511 keV gamma photons emitted in opposite directions until they reach and burst into the scintillation detectors, which are the most common detection elements in PET scanners due to their good stopping efficiency and energy resolution [96,97].

As illustrated in Figure 14, the detection system consists of an array of suitable scintillation crystals coupled to visible light photodetectors, such as photomultiplier tubes or semiconductor-based photodiodes [97].

The reconstruction of an image depends on the successful coincidence or simultaneous detection of the pair of emitted photons since photons that do not arrive in pairs are ignored. Because gamma photons are emitted at nearly 180° from each other, their origin can be inferred through their corresponding straight line of coincidence (or line of response) and the time resolution of the detector [96].

This advanced and expensive medical technique is applied, to cite a couple of examples, in the measurement of blood flow that oxygenates tumour cells and in the measurement of cerebral blood flow [86].

### 3.12. Lab-on-a-Chip and Organ-on-a-Chip

Lab-on-a-chip (LOC) devices, and the more recent organ-on-a-chip (OOC) approach, are advanced in vitro techniques that achieve a highly precise control of fluids on a microscopic scale, making it possible to mimic specific biological processes under a controlled environment [36].

According to [98], 92% of drugs under development will never become commercial due to failures in human clinical trials, even though their research and development take an average of 13.5 years at a cost of USD 2.5 billion. In addition, ethical concerns about the use and sacrifice of animals in the development of these new pharmaceuticals are now of primary interest among the scientific community, as well as the fact that there are logical differences between human and animal biology, which means that findings obtained from animal trials are not necessarily suitable for medical applications [99].

LOC and OOC have been outstanding alternatives to overcome the aforementioned problems within the pharmaceutical industry, allowing for the development of biomimetic micromodels of complete biological structures to test the toxicity and/or efficacy of chemical compounds in vitro, thus helping to accurately predict the results of future clinical trials in humans. An example of such novel systems is the Quasi-Vivo^®^ platform developed by Kirkstall LTD [100,101]. This organ-on-a-plate system has demonstrated successful repeatability across multiple laboratories, accounting for a user base of more than 70 universities [99].

The design and manufacture of this type of device are not trivial; they require very precise processes to ensure the high quality necessary to allow repeatable experimental conditions. Photolithography and soft lithographic techniques are often used to fabricate microchannels [36,102]. In photolithography, a very precise, three-dimensional micropattern (on the order of microns to tens of microns) is transferred onto a substrate with selective exposure to light; then, through soft lithography, these micropatterned substrates are used as casts to produce polymer-based microfluidic devices [102].

In Figure 15, a lab-on-a-chip configuration manufactured by [103] in Russia is presented, consisting of a microfluidic chip designed to dilute reagents in a precisely determined proportion for subsequent mixing. It is intended to cover common medical tasks, such as mixing patient blood with reagents to detect diseases or verify patient compatibility with certain drugs.

On the other hand, an example of an organ-on-a-chip device is shown in Figure 16. Here, ref. [104] developed a lung-on-a-chip to accurately model human lungs for drug screening and toxicology applications.

A porous membrane placed between two chambers was lined on each side with two different cell types: an alveolar epithelium on one side and a capillary endothelium on the other. By applying a vacuum to two additional lateral microchambers, an elastic deformation is produced in the membrane and its two layers of tissue. Once the vacuum is released, the membrane and attached cells relax to their original size, thus replicating the dynamic mechanical distortion of the alveolar–capillary interface that is generated during the breathing process [105], as represented in Figure 17.

A similar approach was used by [106] to mimic peristaltic motion and intraluminal fluid flow in a gut-on-a-chip.

### 3.13. Implantable Sensors

Implantable sensors comprise several implantable devices, such as the cardiac pacemaker [107,108], the implantable cardioverter-defibrillator (ICD) and similar [109,110,111,112,113] continuous glucose monitoring systems [114], and the artificial kidney [115], among others [116].

An implantable sensor can consist either of a single monolithic silicon chip or an assembly of several components, such as substrates fabricated from glass, silicon, ceramic, flexible and rigid polymers, etc.; electronic components, such as integrated circuits, resistors, diodes, capacitors, batteries, etc.; a wide variety of sensors, such as micro-electromechanical systems (MEMS), thermometers, electrochemical elements, etc.; wires for interconnecting elements, making electrode contacts, building antennas and coils, etc.; water getters to keep the internal components dry, such as pastes, granules, thick films, etc.; and, of course, biocompatible packaging [117].

As most of these kinds of systems are implanted directly in soft tissues (such as internal organs, muscles, nerves, arteries, etc.), biocompatibility is a major concern. ISO 10993 (“Biological evaluation of medical devices”) and ISO 14971 (“Application of risk management to medical devices”) are the international standards to follow in this regard. Any foreign body that is voluntary or involuntary introduced into the body will cause the so-called “foreign body reaction”. In the case of implantable devices, the strength of this immune response will depend not only on the materials of its components but also on the size and morphology of the implant itself, which sometimes cause the dominant reaction in the tissue. Due to bioincompatibility issues, most implantable sensors cannot be used for more than a few hours or days [118].

For implantable sensors aimed at dealing with body fluids, hydrodynamic aspects also must be taken into consideration; e.g., an implant designed to be placed directly in a blood vessel should not affect the blood flow [117].

Devices such as the mentioned cardiac pacemaker and ICD are used to monitor the hemodynamic aspects of the blood (such as its velocity) when high precision is required. Long-term stability is also achieved along with sensitivity to heart rhythm disorders [64]. According to [119], ~200,000 permanent pacemakers are implanted in the United States annually and ~1,000,000 worldwide.

Although traditional transvenous cardiac pacemakers (TVP) are a well-established treatment for bradyarrhythmia, the obvious invasiveness involved in its use is commonly associated with complications such as upper extremity deep vein thrombosis, pneumothorax, cardiac perforation with a risk of tamponade, central vein obstruction, and tricuspid valve regurgitation, among others [119,120]. Recently, the leadless cardiac pacemaker has emerged as a less invasive alternative consisting of a small package containing a generator and an electrode system. This device can be implanted in the right ventricle via a percutaneously inserted femoral venous catheter [121].

The ICD is recognized today as an effective treatment for people considered to be at risk of sudden cardiac death; however, similarly to TVP, its implantation carries an unavoidable risk of complications [122].

For patients requiring rigorous monitoring of their glucose levels, implantable systems such as the Dexcom G5^®^ [123], the Medtronic Enlite^®^ [124], and the Abbott Freestyle Libre^®^ [125], are now commercially available. These systems use the catalysed oxidation of glucose caused by the enzyme glucose oxidase (Gox), and their implantation is transcutaneous; therefore, they do not measure glucose directly in the blood, but rather in the interstitial fluid of the subcutaneous tissue [118].

Flexible implants, such as the Cook Medical Cook–Swartz Doppler Probe^®^ for blood flow sensing, are a new trend due to the emergence of novel biomaterials. The Cook–Swartz Doppler Probe^®^ allows for the qualitative monitoring of both arterial and venous blood flow in real-time after a transplant surgery [126,127,128] or vascular anastomosis of free flaps [129,130,131], as is illustrated in Figure 18. The probe consists of a silicon sleeve that is wrapped around the vein distal to the anastomosis without disturbing the blood flow. Metal microclips are used to secure the sleeve.

### 3.14. Future Trends in Biofluids Flow Sensing for Medical Applications

The techniques described in the previous subsections illustrated the past and present of body fluid flow sensing. Fortunately for human health, most of this information is expected to be dated in the short/medium term, as impressive improvements in medicine are constantly emerging.

Papers describing a wide variety of new methods and sensors intended to help clinicians to make more accurate diagnoses and prognoses in diseases that disrupt the normal flow of body fluids are published regularly. Although presenting these promising techniques is beyond the scope of this review, we conclude this section with a brief comment on some of the major future trends that biofluidics in medicine is heading towards.

The perspectives obtained through the information presented in the last two subsections (lab-on-a-chip and organ-on-a-chip devices and implantable sensors) allow us to build a bridge toward highly anticipated improvements that are expected to have an enormous impact on biofluidics for healthcare. The miniaturization and biocompatibility of a new generation of similar technologies will be a step forward toward innovative devices that will either replace biological functions within a system or organ or allow for a continuous real-time analysis of physiological variables, creating personalized medical treatments and solutions to current diseases. Recent advances in this regard are attracting enormous amounts of interest in the scientific community, as can be confirmed in [132,133,134,135,136].

Following this context, three topics deserve special attention: clinical point-of-care (POC), organic electronics, and personalized/precision medicine, as well as the close relationship that these paradigms have with each other.

Commercially available POC devices are today a reality, mainly for glucose sensing (representing 85% of the market). They consist of wearable transdermal patch platforms that measure glucose in the interstitial fluid, allowing continuous glucose monitoring for up to 14 days, thus significantly improving the quality of life of diabetic patients [137].

In general, POC devices can be either attached to the skin (cutaneous) in the form of watches, wristbands, flexible tattoo-based biomedical devices or headbands, subcutaneously, or as complete implantable systems. Their goal is to allow for the real-time monitoring of different variables in various body fluids, offering (ideally) a compact design, long-term stability, and integration with home electronic devices such as mobile phones, tablets, and other similar electronic readers. These features are especially valuable when access to clinical screening tools is poor, as is the case in many developing countries [137].

On the other hand, organic electronics point toward the use of carbon-based semiconductors in electronic and optoelectronic devices [138]. As POC devices nowadays mainly rely on silicon-based electronics, organic semiconductors are the long-awaited next step in this field due to characteristics such as biocompatibility, biodegradability, and mechanical flexibility [137,139].

Organic electronic materials conduct ionic as well as electronic charges that make it possible to open a new channel of communication with living systems [138]. In addition, sensors based on organic semiconductors can be customized for use in applications where the selective sensing of biochemical molecules allows for the tailored treatment of a specific patient or certain groups of patients with common characteristics, offering tremendous potential to change the current paradigm of standard tests and treatments toward personalized healthcare and precision medicine [137].

Biocompatible/biodegradable MEMS based on organic electronics are also receiving increasing attention due to their capacity to transduce physical, chemical, or biological events into electrical signals, which can then be relayed to the outside world through electronic interfaces. Implantable MEMS sensors have been tested, for example, to monitor blood pressure, glaucoma, the vestibular system, pH in blood and tissue, in vivo drug delivery, and urinary bladder pressure (a very promising application since urinary incontinence affects more than half of people over 60 years of age) [117].

The cross-fertilization between organic electronics and clinical POC will, therefore, contribute to making precision medicine a reality, representing enormous opportunities for new interdisciplinary research and leading to new biosensors for medical applications in microfluidics, impacting, indeed, healthcare systems and the quality of life of people.

## 4. Discussion

In this review, we compiled key information aimed at those interested in becoming involved in the broad field of flow sensing for medical applications, regardless of their current branch of research. In the first part, we described physiological aspects related to some of the main body fluids that can be studied for the diagnosis of diseases that cause changes in the normal flow of these biofluids. This part is just as important as the second part, where some of the flow sensing techniques already adopted in healthcare settings were addressed. We did not intend to present an exhaustive review of previous work related to bio-fluidics or state-of-the-art flow sensing techniques for general purposes. In fact, we consider this work to be a broad initial perspective on the biomedical flow sensing field with an interdisciplinary objective, where life sciences researchers, such as biologists, physiologists, or physicians, can be introduced to the technical aspects of flow detection, and engineers, physicists, and technicians can gain an overview of the physiological characteristics of some of the most important biofluids.

This is the reason why we did not deal with concepts such as specificity or sensitivity for the techniques described here, since we do not refer (in most cases) to studies under controlled conditions and very specific applications, but rather to a general description of the principle of operation behind well-established medical techniques. In the same regard, our intention is far from comparing each technique to the others, nor suggesting which one is more suitable than the others for a given purpose since we do not refer, again, to the use of the techniques described here for the diagnosis of very specific diseases; e.g., comparing plethysmography to an organ-on-chip device has no practical implications.

## 5. Conclusions

The development of sensors and devices for biomedical applications is a continuously growing field that requires a multidisciplinary effort. The present content, summarized from various sources specialized in the topics discussed here, aims to fill the gap between physiological and technical knowledge that researchers from different areas often face in the initial stages of studies related to this broad topic.

To accomplish the purpose mentioned above, this paper was divided into two sections covering very different concepts, both of which are equally important in biofluid flow sensing. First, we presented a general description of the key physiological aspects of the most important body fluids, as well as the organs and systems where these biofluids have an impact on their proper functioning. In addition, we briefly mentioned some examples of medical conditions that can affect the described biofluids.

We addressed the structure and function of macrocirculation and microcirculation within the circulatory system, as well as the influence of the lymphatic system on blood circulation, which are the most extensive and detailed topics due to their primary importance. In addition, we described blood flow in the respiratory system, digestive system, brain, and eye. Key aspects of the urinary system, fluids within the gastrointestinal tract, cerebrospinal fluid, serous body fluids, synovial fluid, and other relevant human fluids were also concisely described.

The second part of this review describes well-established flow sensing techniques that are already being applied in health care. We described the basics of angiography, lymphoscintigraphy, indocyanine green lymphography, plethysmography, photoplethysmography, particle imaging velocimetry, ultrasonic/acoustic flowmeters, the laser Doppler flowmeter, the electromagnetic flowmeter, the nuclear magnetic resonance tissue flowmeter, positron emission tomography, lab-on-chip and organ-on-chip devices, and implantable sensors, and, finally, we highlighted some foreseeable trends on the matter.

The existing literature on the topics covered in this review is very extensive, making it impossible to summarize all the knowledge surrounding biomedical flow sensing; however, the purpose of this paper was to compile key information on the aspects of this field so that each researcher can obtain a basis on which to deepen their knowledge according to their own scientific needs.

## Figures and Tables

**Figure 1 sensors-22-06836-f001:**
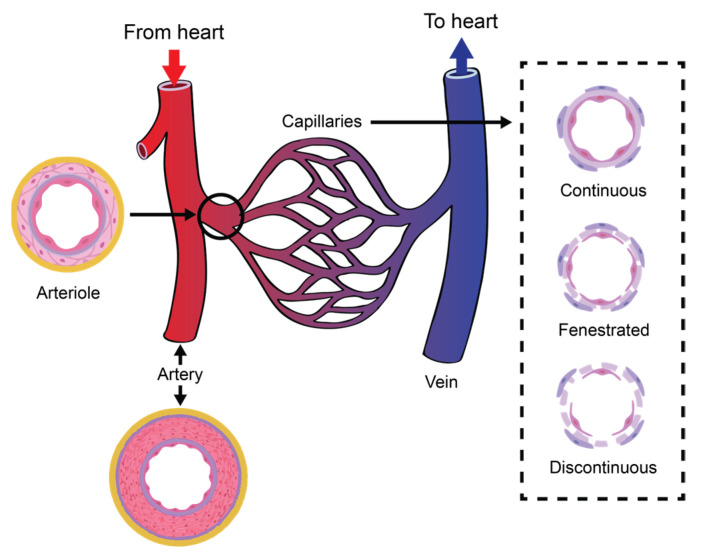
Illustration of a typical vascular tree based on [37]. Arteries and arterioles in blue; capillary beds, veins, and venules in red. All vessels are made of an inner layer of endothelium and an outer layer of the basement membrane. Arterioles and venules are also covered by a second layer of smooth-muscle cells, as well as elastin and collagen fibres. A basement membrane is composed of pericytes and endothelial cells wrapped around capillaries, and this coverage can be continuous, fenestrated, or discontinuous.

**Figure 2 sensors-22-06836-f002:**
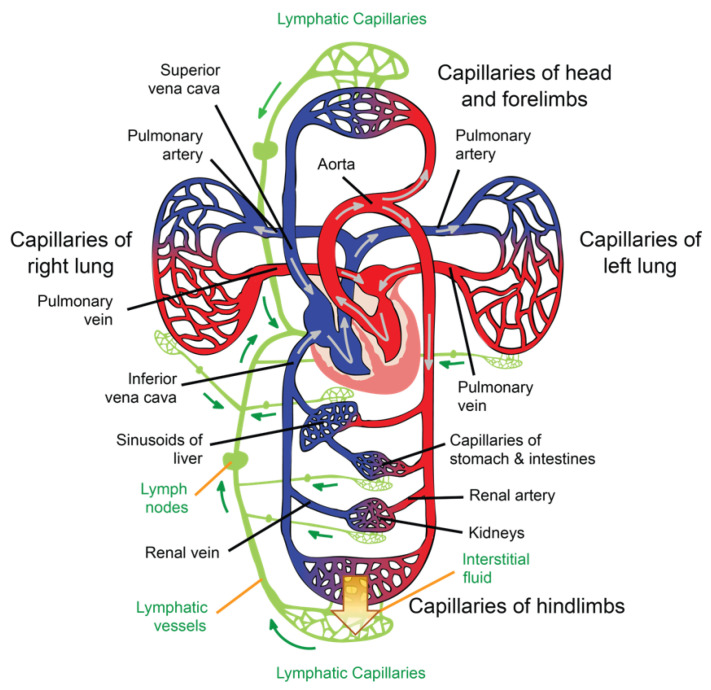
Illustration of the circulatory and lymphatic systems.

**Figure 3 sensors-22-06836-f003:**
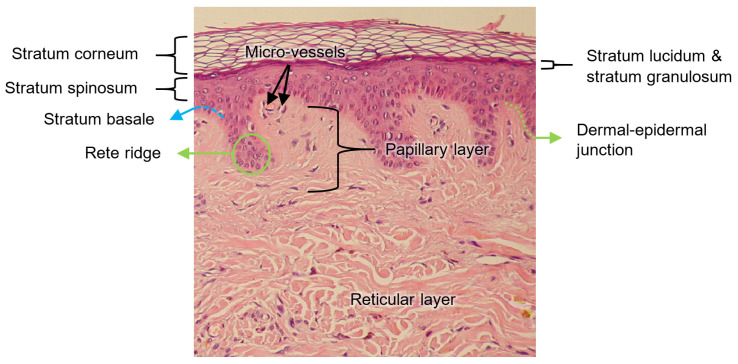
Illustration of the different layers that compose the human skin. Based on [59].

**Figure 4 sensors-22-06836-f004:**
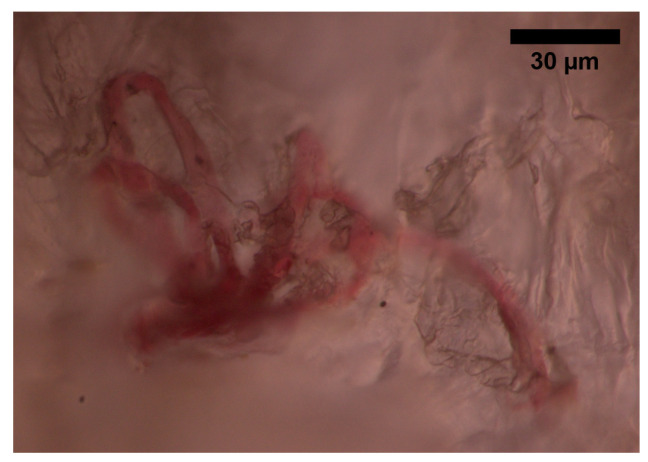
Image of a capillary loop from a sample of pig skin obtained with an optical microscope and an M-Plan APO 50 × 0.55 NA microscope objective.

**Figure 5 sensors-22-06836-f005:**
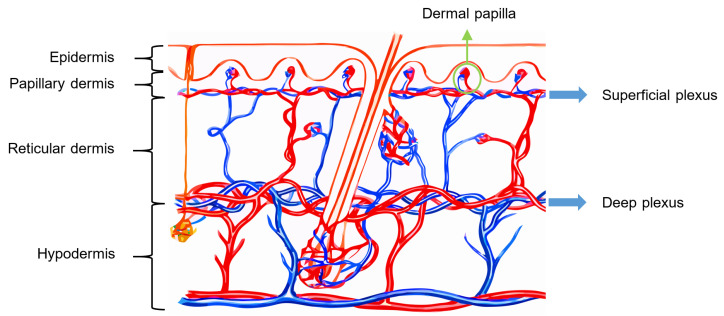
Scheme of the vascularization of the skin. Based in [62,63].

**Figure 6 sensors-22-06836-f006:**
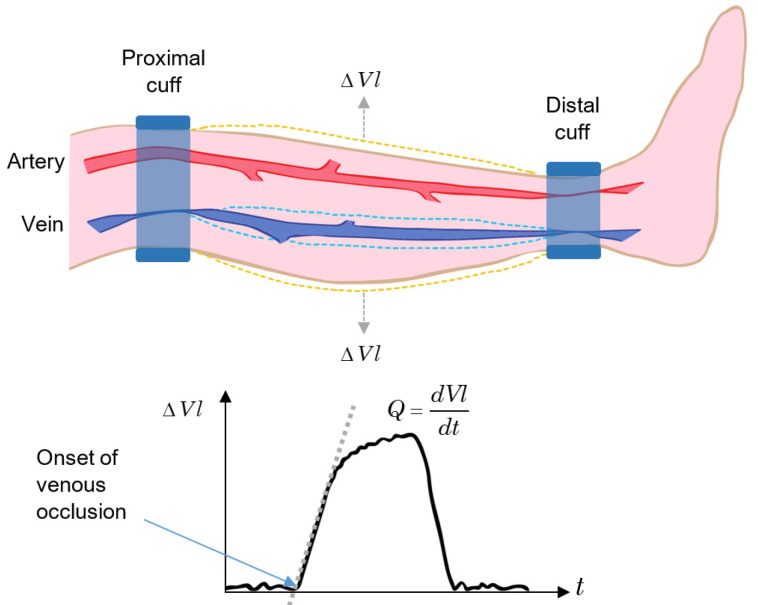
Schematic of the venous occlusion plethysmography method based on [80].

**Figure 7 sensors-22-06836-f007:**
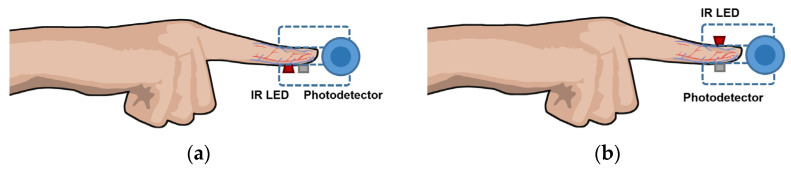
Illustration of (**a**) PPG in reflectance mode and (**b**) PPG in transmission mode. Based on [85].

**Figure 8 sensors-22-06836-f008:**
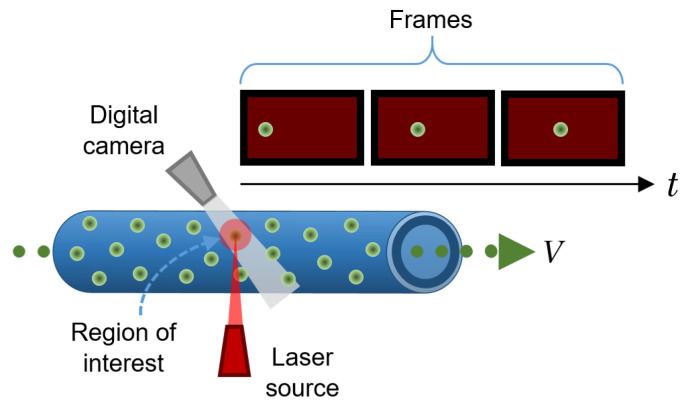
Schematic of a typical PIV setup.

**Figure 9 sensors-22-06836-f009:**
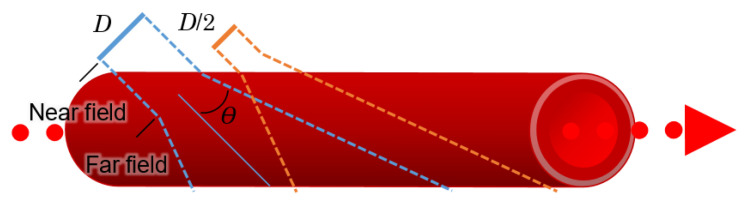
Representation of two ultrasonic transducers with diameters *D* and *D*/2 emitting at the same frequency. As is shown, the difference in diameters causes different fields of penetration.

**Figure 10 sensors-22-06836-f010:**
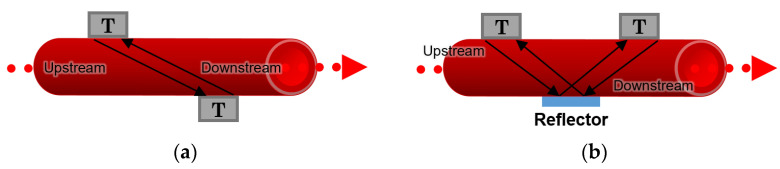
Illustration of both schemes of transit-time flowmeters: (**a**) transducers (T) placed diagonally; (**b**) transducers on one side with a reflector on the opposite side of the blood vessel or artificial channel.

**Figure 11 sensors-22-06836-f011:**
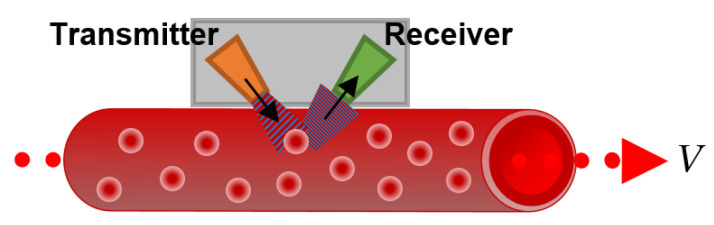
Schematic of a continuous-wave Doppler flowmeter.

**Figure 12 sensors-22-06836-f012:**
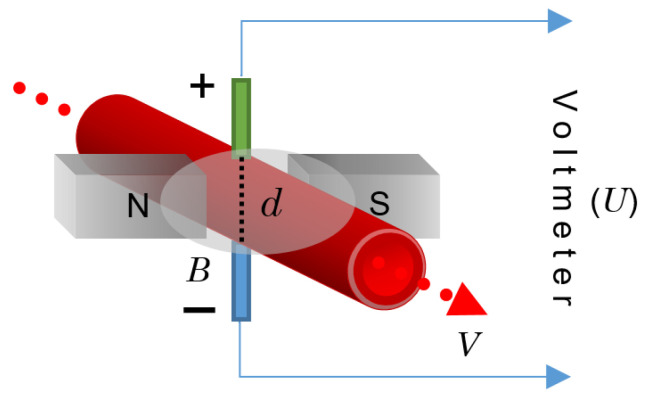
Diagram of an electromagnetic flowmeter: *V* represents the blood flow velocity, *U* is the electromotive force, *B* is the magnetic flux density, and *d* is the channel diameter.

**Figure 13 sensors-22-06836-f013:**
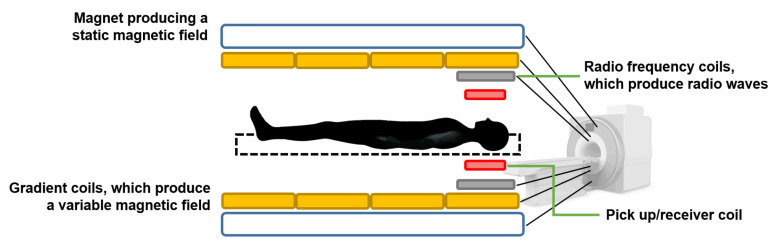
Illustration of a nuclear magnetic resonance tissue flowmeter.

**Figure 14 sensors-22-06836-f014:**
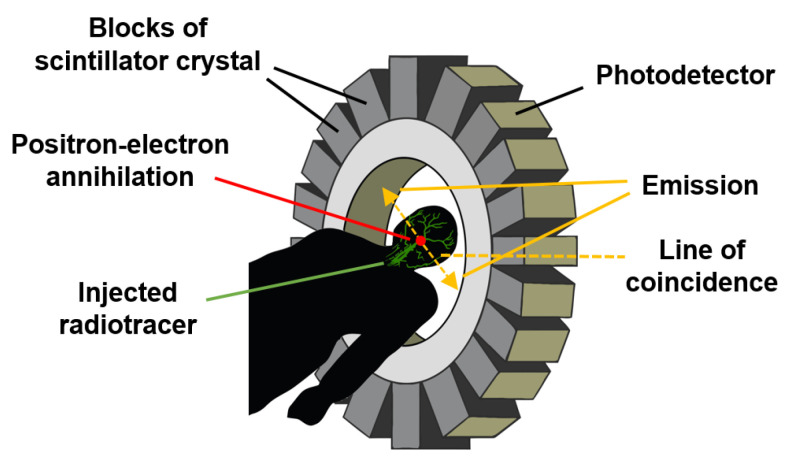
Diagram of a positron emission tomography scanner.

**Figure 15 sensors-22-06836-f015:**
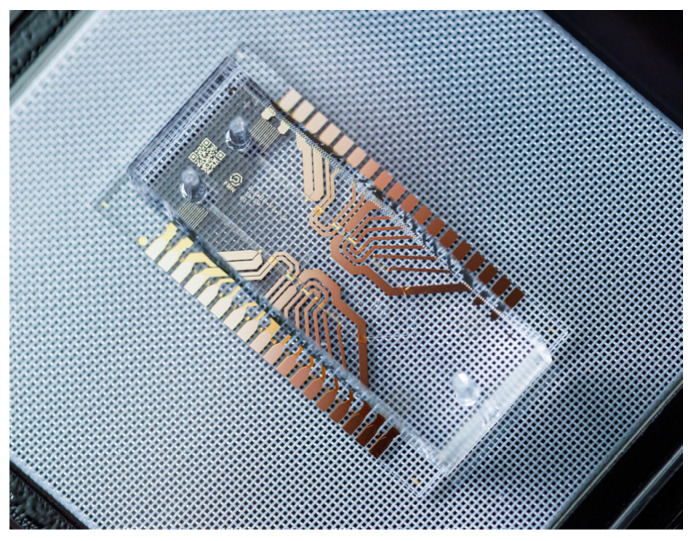
Lab-on-a-chip device for microfluidic applications. The dimensions of the channels are 50 μm high × 200 μm wide. The detection system is made up of metal electrodes on a glass substrate [103].

**Figure 16 sensors-22-06836-f016:**
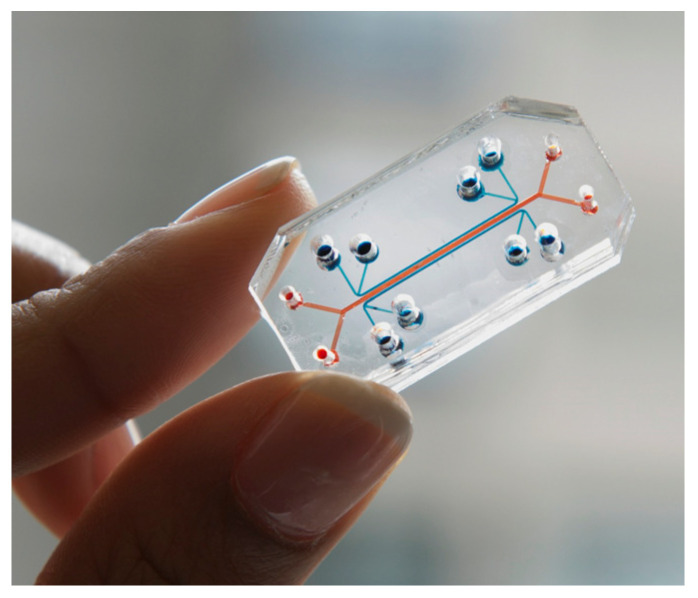
Lung-on-a-chip device developed by [104] to test drugs.

**Figure 17 sensors-22-06836-f017:**
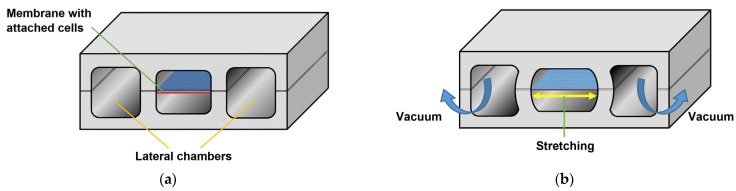
Schematic of the lung-on-a-chip developed by [104,105]: (**a**) in initial relaxed state; (**b**) by applying vacuum to the lateral microchambers, an elastic deformation in the membrane attached with alveolar epithelium and capillary endothelium cells mimics the mechanical distortion of the alveolar–capillary interface produced during the breathing process.

**Figure 18 sensors-22-06836-f018:**
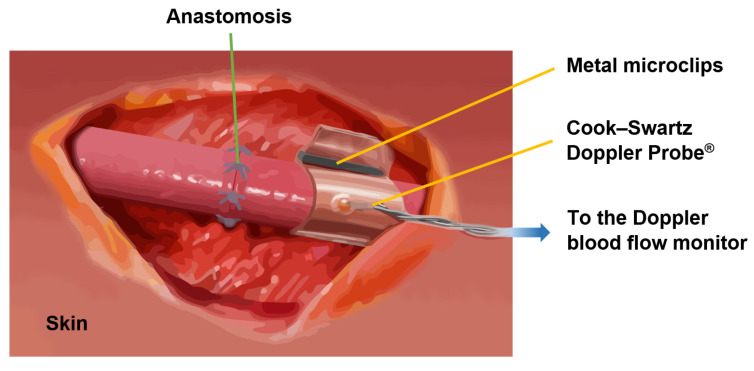
Illustration of the use of the Cook–Swartz Doppler Probe^®^ in the monitoring of blood flow after a vascular anastomosis of free flaps.

**Table 1 sensors-22-06836-t001:** Average dimensions and approximate quantifications of vessels in the human circulatory system for a 30-year-old male with a mass of 70 kg. Adapted from [34].

Vessel	Diameter (mm)	Length (mm)	Number	Total Surface Area (mm^2^)
Aorta	25.0	400	1	31,400
Large arteries	6.5	200	40	163,000
Main artery branches	2.4	100	500	377,000
Terminal artery branches	1.2	10	11,000	415,000
Arterioles	0.1	2	4,500,000	2,800,000
Capillaries	0.008	1	19,000,000,000	298,000,000
Venules	0.15	2	10,000,000	9,400,000
Terminal venules	1.5	10	11,000	518,000
Main venous branches	5.0	100	500	785,000
Large veins	14.0	200	40	352,000
Vena cava ^1^	30.0	400	1	37,700

^1^ Including superior and inferior vena cava.

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
