# Peer review of "Overview of Biofluids and Flow Sensing Techniques Applied in Clinical Practice"

_sensors, 2022, doi:10.3390/s22186836_

Round 1

Reviewer 1 Report

The article “Overview on Biofluids and Flow Sensing Techniques Applied in Clinical Practice”, a review, has its scope well defined and it is mainly divided in two sections: the first related to how main body (blood, urine, serous, synovial) fluids influence human health, and the second a summary of in vivo and in vitro flow sensing biosensors used in clinical context.

This reviews presents to the Sensor readers, essentially focused on the engineering/technological development of biosensors the opportunity to have an introductory view of the biology involved when analysing human body fluids.  This reviewer founds this perspective interesting. I would recomend the authors to address the following:

Section 1

Concerning figures:

1.       Figure 2  and Figure 6 – Did the authors draw it or the picture is taken from a reference? If is taken from a reference please name it in the caption of the figure

2.       Figure 3  and Figure 5– Same comment of figure 2, plus what is the scale of the image, which technique was used in the image acquisition?

3As a general comment, the relevance given to the different human body fluids is different. What was the authors criterion for this discrepancy? It is not clear while reading the manuscript.

Section 2

As a general comment, the relevance given to the different techniques is different. On ones we have schemes, on others we don’t. On ones we have mathematical expressions explaining the physics behind it, on other we don’t, on ones we have explained constrains on other we don’t, and above all what concerns more is that the vast majority is based on the same reference(s).  

1.       Please revise your references of the tecnhiques: or to original papers of the techniques or relevant reviews for each of the technique explained. I acknowledge the authors advocated a concise review, but in my opinion in a review as much as concise it is, a common reference for a book is scarce...  Also there are techniques for example in section 3.1 section 3.8 section 3.10 that only present one reference... for a review I find it scarce and  to be modified.

2.    To maintain consistency, please present schemes for every technique, as for the Sensors reader this could be interesting.

3.       Draw a table where is presented a summary of the techniques, human body fluids in analysis and the respective reference.

4.       Section 3.9 Are the techniques used in clinical context? There is a vast majority on lab on a chip literature for prototypes being tested with high TRL but that are not actually regulated as medical devices and consequently not used in daily clinical context. Are the ones presented used in clinical context? Why those presented are particulary relevant? Why have the authors chosen to present a photograph in here instead of a scheme?

 Please adress the future directions on the analysis of human body fluids.

Author Response

Dear reviewer,

Please find our responses to your kind comments in the attached Word document.

Reviewer 2 Report

The authors described "Overview on Biofluids and Flow Sensing Techniques Applied in Clinical Practice" in literature review syle.

Indeed, as knowledge about the influence of biofluids on human health is of critical importance, this topic should be attractive for potential readers.

I have some recommendations to improve this manuscript.

1. How about lymphatic circulation? Lymph fluid have become a focus of interest for many researchers and clinicians, because there have been few reports of potential roles for lymphatic function. However, I believe lymphatic circulation should be added to this manuscript.

2. Why did you use Figure 3? Figure 3 have no relation to biofluids. Figure 4 is, too.

3. How about bioimpedance spectroscopy and calculating water displacement for the detect of lymphatic disorders? Please add.

  •  

Author Response

Dear Reviewer,

Before anything else, we would like to thank you for your time in reviewing our manuscript and your kind comments about our proposal on the topic of biofluids and flow sensing techniques in healthcare.

Regarding your very interesting observations:

1.- “How about lymphatic circulation? Lymph fluid have become a focus of interest for many researchers and clinicians, because there have been few reports of potential roles for lymphatic function. However, I believe lymphatic circulation should be added to this manuscript.”

This is indeed a very good observation. In our original manuscript we just mentioned the role of the lymphatic vessels in the context of the circulatory system. Now we have written a subsection entirely devoted to the lymphatic system. Also, we have modified Figure 2 to illustrate the function of the lymphatic system draining interstitial fluid.

2.- “Why did you use Figure 3? Figure 3 have no relation to biofluids. Figure 4 is, too.”

In Figure 3, our intention was to illustrate the physiological barriers that any sensor has to surpass in order to reach biological channels inside the human body. We have now removed Figure 3.

Figure 4 (now Figure 3) is maintained in the manuscript as we believe that it adequately illustrates the point discussed above regarding the physiological barriers for flow sensors and, also, it could be helpful for the readers interested in get involved in the development of sensors to measure perfusion or detect diseases involving changes in the flow profile of blood in the skin, such as melanoma.

3.- “How about bioimpedance spectroscopy and calculating water displacement for the detect of lymphatic disorders? Please add.”

Calculating water displacement is a variant of plethysmography, called displacement plethysmography, which uses either water or air-filled containers to measure the change of the volume in the limb. This technique was already mentioned in the plethysmography subsection, however, now we added also a reference to its application in the diagnosis of lymphoedema.

Regarding bioimpedance spectroscopy, as far as we know, it still being a very promissory technique for the diagnosis of lymphatic disorders, however, its clinical application has not been adopted yet. Instead, we added new subsections about lymphoscintigraphy and indocyanine green lymphography, as currently adopted techniques for detecting lymphoedema.

Reviewer 3 Report

This paper is reviewing biofluids and flow sensing techniques. It started with the introduction of biofluids and then moved to the individual techniques sections. However, in my humble opinion, the authors did not provide overall pictures and a comparison of individual techniques. There are no tables or graphical images to summarize all of these techniques. What are pros, cons, capability and limitations for each technique? (Few numbers were presented like for sensitivity, specificity, time etc.) Which technique readers should choose? Technique A vs Technique B? 

Overall, I believe this review not only lacks a big picture summarizing existing techniques but also lacks details in each techniques. As such, the manuscript has great room for improvement.

Author Response

Dear Reviewer,

Before commenting your recommendations, we would like to thank you the time in reviewing our manuscript.

To satisfactorily answer your kind recommendations, we would like first to clarify that the main subject of our manuscript (as we mention in the introduction) is to provide concise information on both, flow sensing techniques and physiological subjects related to biofluids, to allow researchers interested in get involved in this very vast topic to understand the influence and characteristics of the main body fluids in the adequate functioning of human organs and systems, as well as the basic theoretical principles behind the functioning of the most common medical techniques already employed in healthcare to study the characteristics of the flow of such fluids. This is, the first part of the manuscript regarding biofluids is not part of the introduction for later describing the flow sensing techniques as the principal topic, on the contrary, this part is equal important to understand the big picture that we are trying to offer to the Sensors readers, which is far from being just a review of flow sensing techniques. Good reviews on this very specific topic have been presented before.

Comparing individual techniques against each other is far of the scope of this manuscript. We are offering the basic theoretical principle of thirteen techniques that, in some cases, are very different each other; some of them cannot be compared with some others as their individual scopes are different (for example, comparing angiography against photoplethysmography has no practical implications).

Explicitly mentioning pros, cons, capabilities and limitations for the described techniques is also far from the scope of this manuscript, because we are not intending to describe the explicit use of a given technique in the diagnosis of a very specific medical condition. As the diagnosis and study of the different diseases that affect the normal flow of body fluids take into account a wide variety of variables, such as turbidity, changes in viscosity, number of cells in a given volume, velocity, changes in the surrounding tissues, among others; pros, cons, capabilities, and limitations greatly depends on such variables. Cover such quantity of information in order to adequately tackle pros, cons, capabilities and limitations for the thirteen flow sensing techniques presented in our manuscript is not possible.

In the same regard, for most of the cases, sensitivity and specificity can be mentioned only in the context of very specific applications of techniques under well-controlled subjects/samples/patients, due to sensitivity and specificity may vary, for example, because of the age, height, weight, previous pathologies, colour of the skin, among much other variables.

We do not have enough authority to suggest to the reader which technique should choose, furthermore, this is totally contrary to the intentions of this manuscript, as this would imply a dangerous limitation in the judgment of new researchers interested in venturing in this broad field. Precisely (as we mention in the conclusions of our manuscript), the aim of this paper is to compile key information on the aspects of this field, so that each researcher can obtain the bases from which going deeper according to their own judgments.

We are taking into account your observations for the improvement our manuscript by citing now reviews where readers can get detailed information regarding each specific technique discussed in our manuscript. Also, we are adding a Discussion section where we explain some of the points clarified above.

Reviewer 4 Report

This a well-written review which, as the authors say, should help researchers venturing into biofluids flow sensing.

Author Response

Dear reviewer.

We really appreciate your kind comments regarding our manuscript and your time reviewing our work. This encourages us to maintain our efforts in the field of biofluidics.

Reviewer 5 Report

The topic of the review is quite original and is possible, as authors claim in the introduction, it is the first dealing the subject. This reviewer find the review well organized and of wide horizons, authors did not limited the discussion to only some themes, but it is clear the effort to be exhaustive. I am not able to comment deeply all the techniques that authors presented in the review. At least 2 suggestions are mandatory from my side: 

1. Par 3.9 on the lab-on-a chip description is too poor and limited respect the overall work performed in the scientific community. I suggest authors to extend the discussion and references; 

2. A new paragraph could be added dealing with the wide world of organic electronics. Much work has been done for the detection of biomolecules in biofluids (saliva, tears, sweat) by exploiting the interaction of organic semiconductive polymers with biofluids with transistors architecture.    

Author Response

Dear Reviewer,

First of all, it is important for us to thank you for your time reviewing our manuscript, as well as for your kind comments regarding our proposal on the topic of biofluids and flow sensing techniques in healthcare.

Regarding your obsevations:

1.- “The lab-on-a chip description is too poor and limited respect the overall work performed in the scientific community. I suggest authors to extend the discussion and references”

We have expanded the information presented in our manuscript regarding this very interesting and broad topic, and we also did the same with other topics, such as implantable sensors, among others.

2.- “A new paragraph could be added dealing with the wide world of organic electronics. Much work has been done for the detection of biomolecules in biofluids (saliva, tears, sweat) by exploiting the interaction of organic semiconductive polymers with biofluids with transistors architecture.”

The aim of this paper is to cover the main physiological aspects related to body fluids, as well as to provide concise information on well-stablished flow sensing techniques that are already used in healthcare clinics/laboratories. Exploring, in addition to this, promising techniques currently under development, is such a huge endeavour that is impossible to complete adequately.

From this perspective, and in order to maintain consistency with the rest of the manuscript, we do not consider it appropriate to include a section on Organic Electronics; because, as far as we can read in the available literature [1], there are no well-established devices in clinics/laboratories that use Organic Electronics, remaining as a very promising field that nevertheless is worth mentioning to readers interested in biofluids. So we are now addressing Organic Electronics in a new section on new trends in biofluid flow sensing in healthcare, emphasizing the promising future of this technique.

[1] Lee et. al., "Recent advances in organic sensors for health self-monitoring systems". J. Mater. Chem. C, 2018,6, 8569-8612. 

https://doi.org/10.1039/C8TC02230E

Round 2

Reviewer 1 Report

Dear authors, I acknowledge the efforts you made for improving the manuscript and appreciate the thoroughly explanation about how it is structured. I do recommend the publication of this manuscript.

Author Response

Once again, we appreciate your time and kind comments, which were very helpful in improving our manuscript.

Science is a joint effort maintained by the entire research community.

Thank you.

Reviewer 2 Report

The authors revised the manuscript precisely.

Author Response

(The authors gave the same response as above.)

Reviewer 3 Report

Authors clarified the target of this review paper. One comment is that 3.12 Lab-on a chip/ organ on a chip part was little different compared with 3.1-3.11, where the sections actually focusing on imaging, sensing. It was rather focusing on modeling the body fluid and using the device for in-vitro research. This section may be put in the discussion section, introduce more citations related to flow sensing methods, or change some stories etc.

Author Response

Dear reviewer,

Thank you for your kind comments.

Lab-on-a-chip and organ-on-a-chip devices are mainly used nowadays for the applications already discussed in the paper, which are mostly in-vitro techniques for mimic specific human functions related to body fluids in artificial microchannels for the study of some diseases and the screening of new drugs. Although they are different from the previous discussed techniques, they are in fact a bridge linking the presented future trends on biofluids flow sensing with the well-stablished explained methods, along with implantable biosensors. E.g., new organic electronic materials are expected to make it possible to fabricate complete implantable systems similar to those described in the LOC and OOC subsection; these systems will allow reaching the milestone of personalized medicine, since they will be designed and fabricated taking into account very specific pharmacological and physiological characteristics of specific individuals or ethnic groups (some references regarding this topic are added now at the beginning of the subsection related to future trends in biofluids flow sensing in order to better clarify this point).

In fact, in the first version of this manuscript we did not present the subsection on future trends in the field, and we simply let the subsections on LOC and OOC, and Implantable sensors to conclude the paper. However, based on some observations similar to the point of view that you are letting us know in this review, we decided to develop the mentioned subsection, to properly highlight how biofluids flow sensing is progressing.

As already mentioned, we clarify this link between LOC, OOC and implantable sensors with organic electronics and personalized medicine with a new paragraph at the beginning of the subsection on future trends, including some references that support this affirmation. We sincerely hope that you will find this improvement satisfactory.